# Monitoring and Leak Diagnostics of Sulfur Hexafluoride and Decomposition Gases from Power Equipment for the Reliability and Safety of Power Grid Operation

Luxi Yang [1,2], Song Wang [3], Chuanmin Chen [1,*], Qiyu Zhang [2], Rabia Sultana [2] and Yinghui Han [2,*]

1   Department of Environmental Science and Engineering, North China Electric Power University, Baoding 071003, China; 220212223019@ncepu.edu.cn

2   College of Resource and Environmental Engineering, University of Chinese Academy of Sciences, Beijing 101408, China; zhangqiyu22@mails.ucas.ac.cn (Q.Z.); rabiasultana741@mails.ucas.ac.cn (R.S.)

3   College of Materials Science and Opto-Electronic Technology, University of Chinese Academy of Sciences, Beijing 101408, China; wangsong19@mails.ucas.ac.cn

\*   Correspondence: 52350852@ncepu.edu.cn (C.C.); hanyinghui@ucas.ac.cn (Y.H.)

**Abstract:** Sulfur hexafluoride ($SF_6$) is a typical fluorine gas with excellent insulation and arc extinguishing properties that has been widely used in large-scale power equipment. The detection of $SF_6$ gas in high-power electrical equipment is a necessary measure to ensure the reliability and safety of power grid operation. A failure of $SF_6$ insulated electrical equipment, such as discharging or overheating conditions, can cause $SF_6$ gas decomposition, resulting in various decomposition products. The decomposed gases inside the equipment decrease the insulating properties and are toxic. The leakage of $SF_6$ can also decrease the insulating properties. Therefore, it is crucial to monitor the leakage of $SF_6$ decomposed gases from electrical equipment. Quantitative testing of decomposition products allows us to assess the insulation state of the equipment, identify internal faults, and maintain the equipment. This review comprehensively introduces the decomposition formation mechanism of $SF_6$ gas and the current detection technology of decomposition products from the aspects of principle and structure, materials, test effect, and practicability. Finally, the development trends of $SF_6$ and decomposition gas detection technology for the reliability and safety of power grid operation are prospected.

**Keywords:** sulfur hexafluoride ($SF_6$); leakage; decomposing; monitoring; diagnostics; detection

## 1. Introduction

Sulfur hexafluoride ($SF_6$) has been very popular in engineering since its first synthesis in the 1900s by Moissan et al. Pure $SF_6$ is non-toxic, odorless, and insulating and has a stable chemistry [1]. In electrical transmission and distribution electrical equipment, the excellent insulation performance and arc extinguishing performance of $SF_6$ have been used; however, it cannot be completely replaced today. $SF_6$ has become the most popular insulating gas in modern high-voltage power system equipment [2]. Since the 1950s, $SF_6$ gas has been used in gas-insulated transformers (GITs), gas-insulated switchgear (GIS), and gas-insulated transmission line (GIL) equipment in national power industry systems to manage the high voltage carried between power stations and customer load centers. The $SF_6$ gas market is expanding year by year, and according to the research, the demand for $SF_6$ in the electrical industry is nearly 80% of its total production capacity [3].

Due to the widespread use of $SF_6$ gas in power systems, the various life cycles of internal gas insulation equipment will inevitably lead to leaks or malfunction. After electrical failure, $SF_6$ gas will dissociate or decompose into by-products of characteristic component gases, such as thionyl fluoride ($SOF_2$), sulfuryl fluoride ($SO_2F_2$), and sulfur dioxide ($SO_2$), thus affecting the overall insulation performance, as shown in Figure 1. In addition to the discharge, the $SF_6$ decomposition may also occur under the action of

the thermal stress [4]. There have been many studies on the effects of arc discharge and spark discharge, small current discharge, and partial discharge (PD) on the performance of gas-insulated equipment, among which PD is considered to cause 85% of GIS faults [5–7]. Therefore, it is crucial to use various technologies to monitor the discharge situation of the detection equipment for the safe and reliable operation of the power system, especially the local discharge detection.

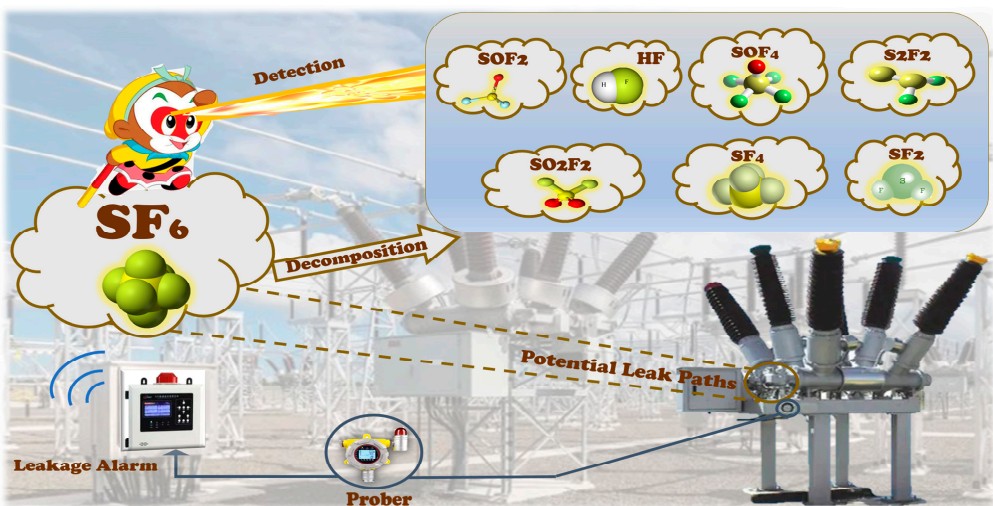

**Figure 1.** Monitoring and leak diagnostics of $SF_6$ and its decomposition gases from power equipment.

Equipment failures can be evaluated by characterizing products to determine the latest monitoring and diagnostic trends in $SF_6$ GIS. In this paper, the decomposition mechanism of $SF_6$ gas and the factors influencing the decomposition of $SF_6$ into by-products are introduced. Some of the latest detection and analysis techniques for $SF_6$ and its decomposition products are introduced, and the structures, performance, advantages, disadvantages, and applicable conditions of various technologies are explained and analyzed. The prominent part of this review is the addition of some recent chemical sensing materials for $SF_6$ decomposition gas identification, which provides a reference for online monitoring technology.

## 2. Decomposition Mechanism of $SF_6$ and Influencing Factors

The purity of the $SF_6$ gas used in the electrical equipment is higher than 99.7%, and the impurities in the equipment are mainly traces of water, oxygen ($O_2$), and sulfur tetrafluoride ($SF_4$). The property of the $SF_6$ gas is very stable. When the temperature is below 150 °C, it is difficult for $SF_6$ to react with other substances, and it does not decompose in the normal working environment. However, in the process of arc, corona discharge, sparks, and overheating (Table 1), $SF_6$, in the case of electron collision or internal heat, will split into a variety of low fluoride sulfide and can be decomposed into low sulfur fluoride, such as sulfur difluoride ($SF_2$) and $SF_4$, which will react with impurities or some materials (electrodes and surface materials of the equipment) in the equipment to produce toxic or corrosive compounds [8–15], such as $SO_2F_2$, $SO_2$, $SOF_2$, and hydrogen sulfide ($H_2S$), as shown in Figure 2. When the fault involves solid insulation materials, carbon tetrafluoride ($CF_4$), carbon monoxide (CO), carbon dioxide ($CO_2$), carbon disulfide ($CS_2$), etc., are also generated [13–15]. In the normal state, the chemical activity of $SF_6$ gas discharge decomposition products is higher than other gases, which are corrosive and toxic. The accumulation of these decomposition gases will corrode the electrode, reduce the insulation performance of the electrical equipment, and affect the normal safe operation of the electrical equipment.

**Table 1.** Characteristics of different discharge types and $SF_6$ gas decomposition products.

| Discharge Type | Discharge Motor Theory | Signal Characteristics | Duration | Discharge Energy/J | Gas Products | Refs. |
|---|---|---|---|---|---|---|
| Arc discharge | Current interruption of circuit breakers, short circuits in gas chambers, and disconnector operation | Temperatures up to 20,000 K, more than a few KA | Tens to hundreds of milliseconds | $10^5$–$10^7$ | $SOF_2$, $SO_2F_2$, $SF_2$, $SF_4$, $S_2F_2$, $SOF$, $SO_2$, $HF$, $CO_2$, $H_2S$, and $COF_2$ | [11,12] |
| Spark discharge | Air gap capacitive discharge, such as breakdown during high-voltage tests | High instantaneous current | Microseconds | $10^{-1}$–$10^2$ | $SF_4$, $S_2F_2$, $SOF_2$, $SOF_4$, $SO_2F_2$, $SO_2$, $S_2O_{10}$, $S_2O_2F_{10}$, and $S_2F_{10}$ | [12,13] |
| Thermal decomposition | When high-frequency current passes through the contact of the oxide layer; electrical fault or short circuit caused by damaged insulation or excessive current load. | 373 K–473 K | - | - | $SF_4$, $SF_2$, $SO_2$, $CS_2$, $SOF_4$, $SO_2F_2$, $H_2S$, $SO_2$, $SOF_2$, $S_2OF_{10}$, and $CO$ | [3,12,14] |
| Corona or partial discharge | Caused by defects on the electrode surface | 10–$10^5$ pC | 0.1–10 ms | $10^{-3}$–$10^{-2}$ | $CF_4$, $SOF_4$, $SO_2$, $SO_2F_2$, $HF$, $SOF_2$, $CS_2$, $H_2S$, and $CO_2$ | [11,13,15] |

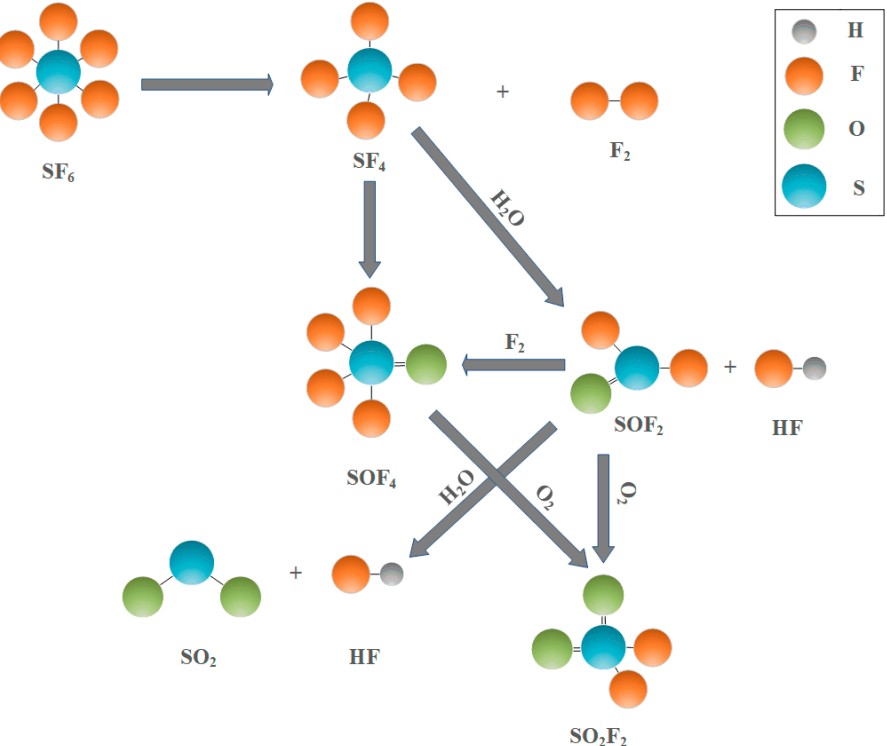

**Figure 2.** $SF_6$ decomposition process diagram.

The concentration and rate of $SF_6$ decomposition characteristic gases are correlated with the insulation fault difference during and at the end of the generation process [16]. Different insulation defects can lead to different compositions of the decomposition gas, and the composition, generation amount, and stage production rate of the corresponding decomposition gas components will also be very different [17]. For example, the characteristic decomposition product of $SF_6$ under superheating conditions is $H_2S$. However, the production of $SF_4$ and $CS_2$ is related to solid insulation failure.

The concentration and generation rates of $SF_6$ decomposition gases are also associated with the concentration of $O_2$ and water in the $SF_6$ buffer gas. Tang et al. found that the higher the $O_2$ concentration was, the more $SOF_2$ and $SO_2F_2$ were produced [11]. In addition, humidity also affects the service life of the insulation equipment. Water vapor has the ability to trap fluorine atoms and inhibit the recombination of low-fluorine sulfides into $SF_6$, which promotes the production of corrosive acid gases. In electrical equipment, water

vapor can be released gradually from the inner surface and electrode material or come from outside in unsealed components. To ensure normal operation, the moisture content of the buffer gas in the equipment shall be controlled below 800 ppmv. Zeng et al. [18] reported that $SF_6$ decomposition characteristics under partial discharge (PD) are not only associated with insulation fault type and its severity but also closely related to the $H_2O$ content in the equipment. Higher humidity caused more favorable generation of $SO_2$ and HF (hydrogen fluoride), while $H_2O$ had no significant effect on $CO_2$, but the water depleted intermediate by-products ($CF_2$ and $CF_3$) and inhibited $CF_4$. These results show that the $O_2$ and water content have some effects on the decomposition products of $SF_6$ and the internal insulation state of the equipment under high pressure.

The degree of insulation damage and the development trend of the insulation situation affect decomposition composition, content, and change in the $SF_6$ gas. Previous experimental studies demonstrated that the ratio of $SO_2$, $SOF_2$, and $SO_2F_2$ is advantageous for determining the degree of discharge in electrical equipment. Under higher-intensity discharge conditions, the volumetric fraction ratio of $SOF_2/SO_2F_2$ and $SO_2$ content increases more rapidly [18]. The discharge decomposition of $SF_6$ involves not only complex physical changes but also complex chemical processes. Defect type, solid insulation material, water content, micro-oxygen original content, discharge voltage, electrode material, and discharge current can affect the final generation [19].

## 3. Monitoring and Detection Technology of $SF_6$ and Its Decomposition Components

The insulation state of the equipment can be monitored and diagnosed by analyzing the leakage and decomposition characteristics of $SF_6$ in power equipment. At present, a variety of gas analysis technologies have been reported, including detection equipment, testing materials, and so on. Monitoring technology can obtain continuous data on various variables in the environment in real time and can be directly applied to environmental management work such as environmental change trend analysis, pollutant emission estimation, environmental quality evaluation, etc., to improve work efficiency and data quality. At the same time, it is a non-interference real-time observation and detection of the environment in its natural state and will not affect the environment itself, thus obtaining more accurate and reliable data. The accuracy of monitoring equipment is generally high, which can more accurately reflect the real situation of environmental variables. In addition, unattended automatic monitoring can be achieved, greatly reducing the input of human and material resources [20].

At present, the mainstream greenhouse gas monitoring technology is based on the interaction of light and gas components as the physical mechanism, according to the characteristic spectrum of the target component, with the help of spectral analysis algorithm, combined with optomechanical-electrical engineering technology, to achieve the non-contact quantitative inversion of greenhouse gas concentration in response to differences in time, space, and distance. The ideal optical method is characterized by a short response time, high accuracy, lack of requirement for calibration, high sensitivity, moderate cost, and the ability to detect a gas in real time. Common greenhouse gas spectroscopy monitoring technologies mainly include non-dispersive infrared spectroscopy (NDIR), Fourier transform spectroscopy (FTIR), differential optical absorption spectroscopy (DOAS), differential absorption lidar (DIAL), tunable diode laser absorption spectroscopy (TDLAS), off-axis integrating cavity output spectroscopy (OA -ICOS), optical cavity ring-down spectroscopy (CRDS), laser heterodyne spectroscopy (LHS), space heterodyne spectroscopy (SHS), photoacoustic spectroscopy (PAS), Raman spectroscopy (RS), silicon photomultiplier method (SiMP), etc. The typical techniques for detecting $SF_6$ gas and its decomposition gases are described in detail below.

### 3.1. Nondispersive Infrared Absorption Spectroscopy (NDIR)

The non-dispersive infrared spectroscopy (NDIR) method is a mid-IR spectroscopic gas monitoring technology, which analyzes gases according to the mid-infrared characteristic

absorption wavelength caused by the vibration of the gas molecules. In the actual operation and production of power equipment, the NDIR sensor has been used to locate the sulfur hexafluoride leakage area through multi-point arrangement, data acquisition, data analysis and processing, and communication technology and to monitor the operation status of equipment at the remote end [21].

### 3.2. Photoacoustic Spectroscopy (PAS)

The PAS technology has an independent excitation wavelength and can detect trace gases because the gases have specific absorption of light excitation sources at different wavelengths. Gas molecules produce sound waves that can be detected through various spectral receivers, such as optical fiber tips, electret microphones, and quartz tuning forks [22]. The principle of absorption spectroscopy of infrared light conforms to the Beer–Lambert law. Because the gas has an absorption effect on the infrared light, the gas will absorb the infrared light, resulting in a decrease in the light intensity. In addition, the higher the gas concentration, the more significant the attenuation of light intensity [23]. Absorbance ($A$) represents the absorption situation of infrared light [24]. The relationship between absorbance $A$ and incident light intensity $I_0$ ($\lambda$), transmitted light intensity $I$ ($\lambda$), medium concentration $C$, and effective optical path length $L$ is as follows:

$$A = \ln[I_0/I(\lambda)] = kLC \tag{1}$$

### 3.2.1. Microphone Photoacoustic Spectroscopy

If the minimum detection limit of $SO_2F_2$ and $SOF_2$ can reach 2 ppmv, then the detection accuracy of GIS can be achieved. Zhang et al. [25] used the MIR (mid-infrared spectroscopy) distributed feedback quantum cascade laser (DFB-QCL) and electronic microphone, and a steel resonant optical sound unit (PAC) was designed and manufactured. $SO_2F_2$ and $SOF_2$ were measured at 6648 nm and 7463 nm. The minimum detection limitation of $SO_2F_2$ and $SOF_2$ is 0.22 ppmv and 0.28 ppmv, respectively, which can meet the precise requirements required for GIS gas detection.

Carbon monoxide (CO), as a trace characteristic component, is usually used to identify superheated insulation defects in GIS equipment. Since leaked $SF_6$ gas and various by-products may be cross-absorbed in the mid-infrared (MIR) region, making it difficult to distinguish, which interferes with the monitoring results and affects identification accuracy. In this case, the accuracy of the detection results does not meet the <1 ppmv detection requirements of the $SF_6$ gas insulation equipment [26,27]. Qiao et al. [28] demonstrated the $SF_6$-enhanced vibration-translational relaxation process in the near-infrared region based on PAS and realized that the minimum detection limit of CO is 467.5 ppb. Yin et al. [16] fabricated a multi-component gas sensor by the near-infrared (NIR) or ultraviolet (UV) spectral region emitting excitation sources and resonant photoacoustic unit (PAC) for online monitoring (Figure 3a), and CO, $H_2S$, and $SO_2$ were detected and quantitatively measured by time-division multiplexing (TDM) methods. The minimum detection limits for CO, $H_2S$, and $SO_2$ were 435 ppbv, 89 ppbv, and 115 ppbv, respectively.

### 3.2.2. Quartz-Enhanced Photoacoustic Spectroscopy (QEPAS) Method

Quartz-enhanced photoacoustic spectroscopy technology has a strong anti-interference ability against external noise and has a small size, low cost, and fast response, making it the first choice for trace gas detection in various complex situations. The quartz tuning fork (QTF) can detect the sound waves generated when the gas absorbs and modulates the light [29]. After the resonant frequency of the laser beam bending mode of the QTF in-plane modulation occurs, the resonant tube produces a standing-wave vibration mode, and the tip deflects. This causes the appearance of electrical signals, with the signal being proportional to the concentration of the absorbed analyte [30]. Laser sources of different wavelengths can detect a wide range of trace gases, from ultraviolet to terahertz, up to parts per billion [31].

The QEPAS measurement system is widely used for transient gas measurement in various complex scenarios. Wang et al. [32] reported a quartz-enhanced photoacoustic $SF_6$ sensor for ventilation studies using a 10.5 μm continuous-wave distributed feedback quantum cascade laser (QCL). The $SF_6$ sensor locks the QCL wavelength at the absorption peak of $SF_6$ by detecting sound waves induced by gas absorption, with a detection limit of 4.6 ppb. This sensor can capture transient changes in $SF_6$ concentration during ventilation. Fu et al. [33] studied a fast response $SF_6$ measurement system based on quantum cascade lasers and compared it to commercial instruments for instantaneous $SF_6$ tracer gas measurement. The QEPAS system can be used for transient tracer gas measurement. With a data acquisition interval of 0.4 s, the QEPAS system can record the $SF_6$ concentration in real time after rapid changes and can successfully capture the concentration peak, while commercial instruments have mostly failed to do this.

The humidity situation determines the degree of failure of the insulated equipment. In order to obtain the humidity content of the monitored equipment and ensure the normal operation of the equipment, Yin et al. [22] used a commercially available near-infrared distributed feedback laser (DFB) to generate feedback and design a gas sensor system using QEPAS technology. According to the linear response of 0.14~2.37% water vapor concentration and optical signal, the fitting degree was good, and there was a linear correspondence to the water content. The final detection limit of the sensor was 0.49 ppmv.

Sun et al. [9] designed a QEPAS gas sensor to detect a trace CO gas in an insulating device (Figure 3b). The center wavelength of the quantum cascade laser was 4.61 μm, and the spectral characteristics of the CO were obtained. The minimum detection limit was able to reach 10 ppb, and the sensor response time was 3 min.

### 3.2.3. Cantilever-Enhanced Photoacoustic Spectroscopy (CEPAS) Method

Studies have shown that cantilever-enhanced photoacoustic spectroscopy has the potential for research in the sensitivity and selectivity of trace gas analyzers, and the detection sensitivity of trace gases can reach the ppt level [34,35]. Karhu et al. [36] used a cantilever to enhance the photoacoustic spectrum of benzene to reach a detection limit of less than 1 ppb. The detection limit of trace acetylene was 80 ppt [37]. The noise equivalent detection limit of HCN and $CH_4$ was able to reach 190 ppt and 65 ppt [38].

Zhao et al. [39] proposed a cantilever-enhanced optical fiber photoacoustic microprobe, using UV multi-mode fiber and single-mode fiber to transmit excitation light and detection light from UV led, respectively, for $SO_2$ detection. Allen–Werle deviation analysis was used to evaluate the long-term stability of the system, with white noise as the system noise, and $SO_2$ detection limits of 35 ppb and 5 ppb in the $SF_6$ and $N_2$ background, respectively.

Cheng et al. [40] designed a $C_2H_2$ and CO detection system based on a microcantilever and single-channel photoacoustic unit. The minimum detection limits for $C_2H_2$ and CO were 0.27 ppmv and 23.42 ppmv, respectively. This system simultaneously detected these multi-component gases. Chen et al. [41] used the distributed feedback semiconductor laser (DFBLD) as the light source to build a cantilever-enhanced photoacoustic spectroscopy detection system and then studied the photoacoustic spectrum of carbon-monoxide at $4291.5 \text{ cm}^{-1}$. The high stability of the photoacoustic signal had a good linear relationship with the gas concentration, and the detection limit of carbon monoxide was able to reach 5.1 ppmv. Cheng et al. [42] studied the relationship between PA signal and CO concentration under $SF_6$ and $N_2$ backgrounds, which was based on a cantilever-enhanced PA detector (Figure 3c). The infrared absorption wavenumber at $6380.318 \text{ cm}^{-1}$ was selected, and the central wavelength was 1567 nm infrared laser. Finally, the minimum detection limit was 3.63 ppmv in the $N_2$ background and 9.88 ppmv in the $SF_6$ background.

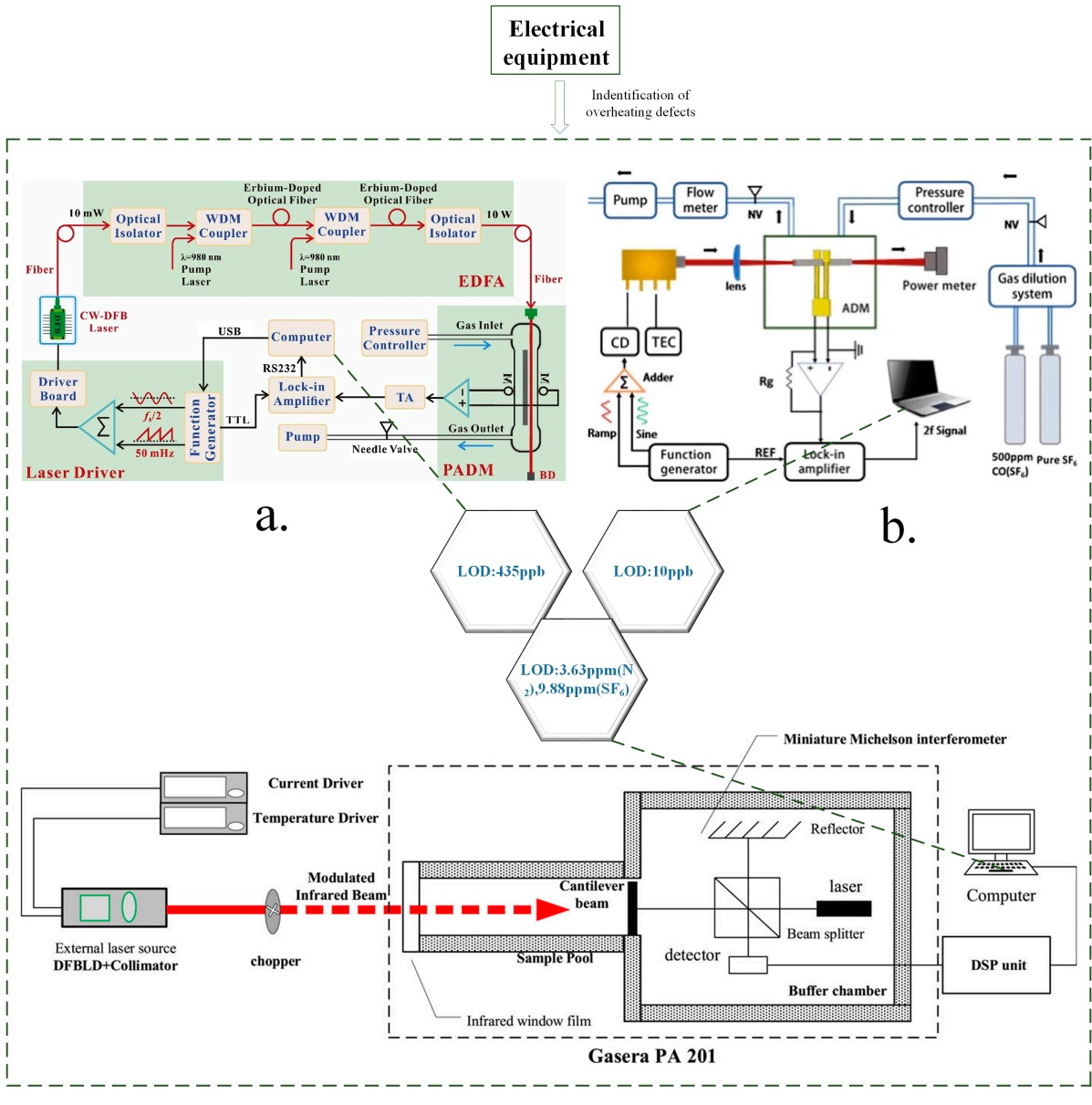

**Figure 3.** Schematic diagram of the carbon monoxide detection laser and the limit of detection. (**a**) Microphone photoacoustic spectroscopy [16]. (**b**) Quartz-enhanced photoacoustic spectroscopy (QEPAS) [9]. (**c**) Cantilever-enhanced photoacoustic spectroscopy (CEPAS). Adapted with permission from [42].

### 3.3. Fourier Transform Infrared Spectroscopy (FTIR) Method

FTIR technology can obtain the infrared absorption spectrum of the measured gas by measuring the interferogram of infrared light and conducting Fourier integral transformation on the interferogram. It can realize the simultaneous monitoring of multiple components and is suitable for the background, profile, space–time change measurement, and isotope detection of greenhouse gases. The instrumentation system is relatively complex and expensive. It also has the advantages of high resolution, fast response, large number of detectable elements, high detection accuracy, strong anti-interference ability, etc., which can achieve qualitative and quantitative detection effects of various gas decomposition components. However, due to the characteristic peaks of $SF_6$ and the proximity of partial decomposition products, the existence of cross-interference is challenging for

engineering applications. FTIR uses the spectral properties of gas molecules to detect gases [43]. Different gases have different infrared absorption spectra. The quantitative detection of infrared absorption spectroscopy of gases is mainly carried out according to the Beer–Lambert law. Zhang et al. [44] used $SF_6$ as a background gas and used FTIR to analyze the characteristics of $SOF_2$, $SO_2F_2$, CO, and $SO_2$ decomposition components. The wavenumbers corresponding to the optimal absorption peaks for the four gases were 530, 544, 2169, and 1167 $cm^{-1}$, respectively (Figure 4a). Figure 4b shows that as the PD time gradually increases, the gas absorption intensity of each component increases. There is a quantitative relationship between gas yield, PD time, and PD amount under five different PD conditions.

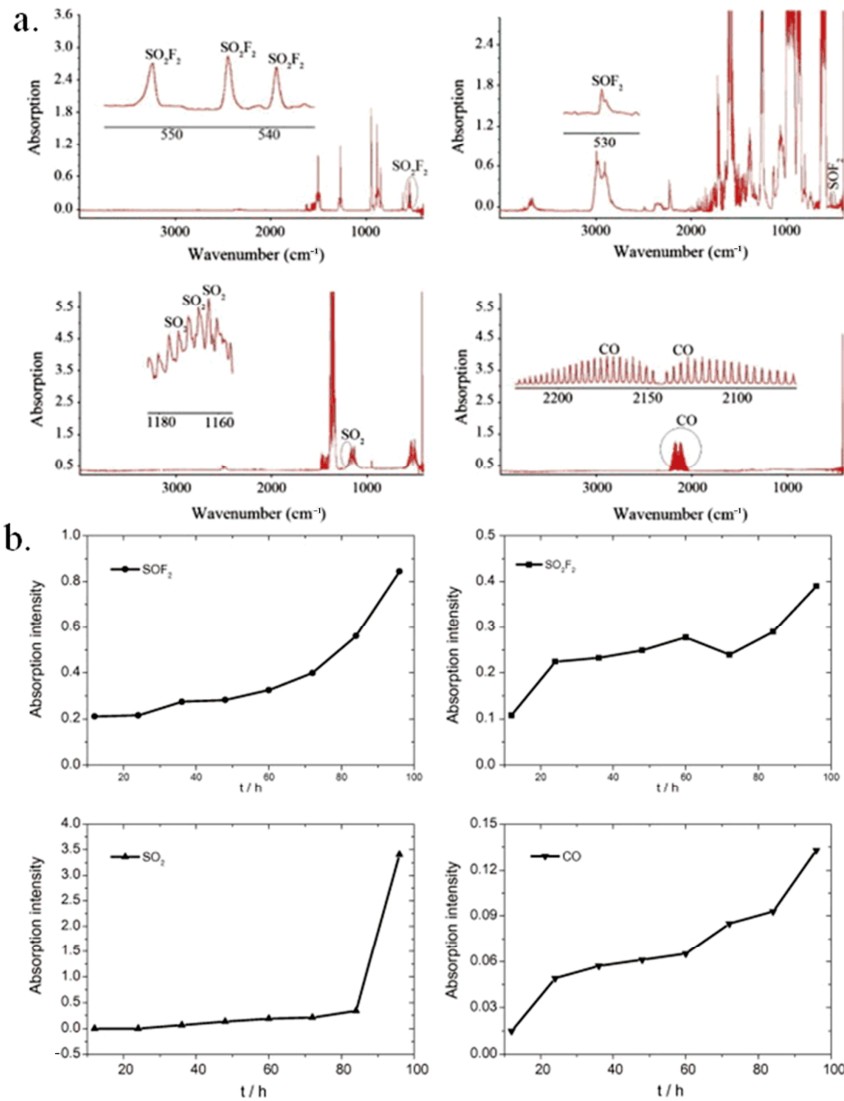

**Figure 4.** Fourier transform infrared spectroscopy quantitative analysis of $SF_6$ partial discharge decomposition components. (**a**) Standard IR spectra of $SO_2F_2$, $SOF_2$, $SO_2$, and CO. (**b**) Absorption intensity of decomposition components with PD time. Adapted with permission from [44].

In order to solve the problem of poor adaptability of traditional quantitative models to nonlinearity, Shi et al. [45] combined Fourier transform infrared spectroscopy (FTIR) with SCARS-DNN (stability competitive adaptive reweighted sampling-deep neural network). The results showed that the lowest detection limits of $SO_2$, $SO_2F_2$, and $CS_2$ were 58.03 ppbv, 55.95 ppbv, and 60.72 ppbv, respectively, under the interference of high $SF_6$ concentration.

The relative error of the three decomposition products is no more than 1.36% and the maximum absolute error is only 0.48 ppm.

### 3.4. Tunable Semiconductor Laser Absorption Spectroscopy (TDLAS) Method

TDLAS technology has been valued for its advantages of fast response speed, high sensitivity, and high spectral resolution, which can realize on-site point-based and regional open greenhouse gas detection, especially for the monitoring and analysis of trace greenhouse gases such as $SF_6$. TDLAS technology uses a tunable laser source with a narrow linewidth to fully scan one or more absorption lines of a gas molecule. If multi-gas component detection is carried out, laser multiplexing is usually required. The theoretical basis of TDLAS technology is the Beer–Lambert Law. Gas concentration is measured by selecting gas molecules in a high-resolution spectrum to absorb laser energy, which attenuates the power of the laser. Then, the absorption attenuation of the laser signal during processing and acquisition is amplified to obtain the gas concentration.

The optical cavity structure serves as the main part of the laser. Zhang et al. [46], based on theoretical analysis, selected optical cavity spectroscopy technology to develop a TDLAS detection system. The detection accuracy of HF is 3 ppmv. To obtain the spectral information in the $SO_2F_2$ and $SOF_2$ detection bands, Zhang et al. [47] established an experimental platform for TDLAS based on mid-infrared spectroscopy. The cross-interference analysis of $SF_6$ characteristic components such as $H_2S$, $SO_2$, CO, and $SF_6$ background gases shows that $SO_2F_2$ can be detected when the output wavenumber is 1506.4 $cm^{-1}$. When the output wavenumber is 1340.2 $cm^{-1}$, the $SOF_2$ result can be detected. When the concentration of $SO_2F_2$ gas is greater than 1.01 and less than 19.88 ppm, the maximum relative error is 2.044%. When the $SOF_2$ gas concentration is greater than 1.03 and less than 9.00 ppm, the maximum relative error is 0.728%, which has a good linear relationship.

### 3.5. Raman Spectroscopy (RS)

Raman spectroscopy is one of the spectroscopic techniques that can be used for gas sensing. Raman spectroscopy uses a single-wavelength laser that enables simultaneous analyses of most gases. However, the Raman scattering cross-section is relatively small, and the Raman signal provided by the gas is very small [48,49]. Fiber-reinforced Raman spectroscopy (FERS) improves the Raman signal of the gas. The gas fills the core of the hollow optical fiber, which allows the optical fiber to be used in a gas battery so that the Raman signal is greatly improved [50,51]. Cavity-enhanced Raman spectroscopy (CERS) can also improve Raman signals, with increased laser power based on optical interference at its core. Multiple-pass cavity, Fabry–Perot cavity, laser cavity, and microcavity are several commonly used, strong enhancement techniques that can improve the Raman scattering intensity and LOD of Raman spectral gas sensors [52].

Cavity-enhanced Raman spectroscopy (CERS) has high sensitivity, selectivity, and accuracy of results and high availability for a variety of gas measurements. Wang et al. [10] designed a dissolved gas analysis (DGA) method using a CERS device (Figure 5a) based on a 110 kV power transformer for equipment diagnosis. $SF_6$ was used as the internal standard gas. CERS equipment extracts and analyzes dissolved gases. Figure 5b,c shows the Raman spectrum of CO and $CO_2$ at 0.5 kPa. The limit of detection (LOD) values for CO (2142 $cm^{-1}$) and $CO_2$ (1388 $cm^{-1}$) were 1.22 Pa and 0.54 Pa, respectively. Finally, the ppmv level sensing was achieved by $N_2$, $O_2$, $CO_2$, CO, and $H_2$ at a total pressure of 1 bar. If the exposure time is more than 10 min, subppm level LOD can also be achieved. The Raman signal obtained by CERS has been shown to have an excellent linear relationship with gas partial pressure, laser power, and exposure time.

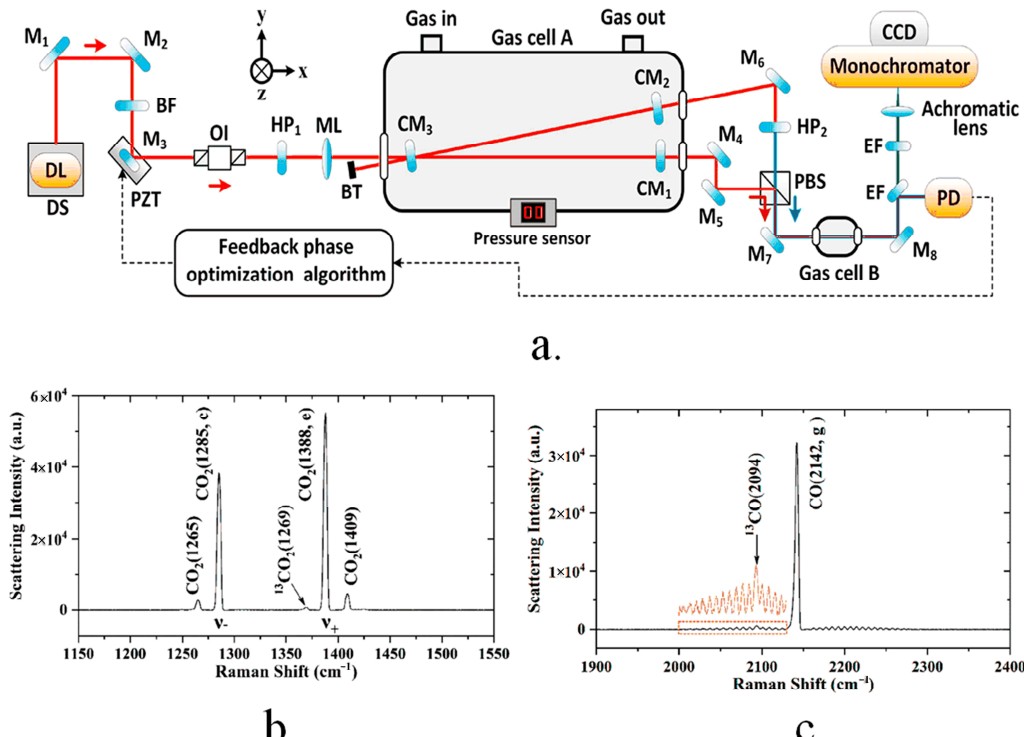

**Figure 5.** (**a**) Experimental setup diagram. (**b**) Raman spectra of 0.5 kPa $^{12}CO_2$ and $^{13}CO_2$. (**c**) Raman spectra of 0.5 kPa CO. Adapted with permission from [10].

Based on density functional theory, experts built a simulated Raman spectrogram, designed a gas sample pool, and set up an experimental platform to study the pretreatment method and spectral signal enhancement technology of Raman spectrogram detection of characteristic components of sulfur hexafluoride gas decomposition and made modifications. The field detection and application research of the characteristic components of sulfur hexafluoride gas decomposition based on Raman spectroscopy were carried out and put into production after repeated use, improvement, and verification.

### 3.6. Gas Chromatography (GC)

Gas chromatography (GC) is a collection of material separation and detection and integration technology produced in the 1950s. GC can detect multiple components with high sensitivity and good quantitative accuracy. At present, it is very popular in the field of $SF_6$ decomposition component detection. Gas chromatography can detect most of the gas components, including simultaneously detecting $SOF_2$, $SO_2F_2$, $SO_2$, and $SF_4$, with a detection accuracy of up to μL/L [53]. However, this method is costly, has a long response time, and requires a large size; in addition, the column must be cleaned regularly. Currently, GC detection is relatively common in the laboratory environment and is not suitable for in-line monitoring or portable testing.

### 3.7. Ion Migration Spectrometry (IMS) Method

Ion mobility spectrometry is a kind of monitoring method for separating and measuring objects according to the difference in ion mobility under environmental pressure. It has the advantages of fast response time, high sensitivity, low cost, and simple operation. IMS has been used to monitor the purity and quality of $SF_6$ in gas-insulated switchgear. The drift tube is filled with sample gas, and the mass of the gas is sensed by the IMS peak displacement caused by various ions formed by impurities.

Huang et al. [54] studied a self-built IMS device that uses drift gas as clean air to monitor $SF_6$ in ambient air in real time. The leakage simulation of $SF_6$ was carried out to control the start-up and leakage of $SF_6$ manually. The final measurement has a temporal

resolution of less than 1 s and a detection limit (LOD) of 0.16–0.68 ppmv. IMS has great application potential in monitoring $SF_6$ concentration in the environment.

### 3.8. Miniaturized Monitoring Method

In recent years, nanotechnology and memory sensors have received extensive research attention. The design of gas sensors has two main categories. The first one is to detect the changes in chemical composition. The target gas reacts with the gas-sensitive film, causing changes in the conductivity of the gas-sensitive film and converting it into voltage or current signals. The second is to detect physical changes. The input electrical signal is converted into a mechanical wave by a sensor, which is then converted into an electrical signal under the action of the device. The gas is measured by the characteristic change between the input and output electrical signals. Measuring the surface wave frequency of the film can obtain the change in the gas concentration. This section is based on recent gas-sensitive materials to describe their detection performance for $SF_6$ and its decomposition gases.

The detection tube method is a method to identify and analyze the decomposition products using chemical reactions occurring in test tubes [55]. The commercial test tube method can quantitatively detect the components of $SO_2$, $H_2S$, $CO$, $CS_2$, and HF, which have strong pertinence and less gas consumption [56]. Although its detection accuracy can reach µL/L, it is susceptible to temperature and humidity, which reduces its stability; cross-interference is inevitable, and the presence of high cross-sensitivity limits the accuracy of detecting gases [57]. In addition, storage time can easily affect the results, and some major decomposition gases (such as $SOF_2$ and $SO_2F_2$) will not be detected. As mentioned above, it does not fully reflect the decomposition gas components of $SF_6$ discharge and can only be used as an auxiliary method. Currently, it is widely used in field testing.

In recent years, nanotechnology and memory sensors have received a lot of attention. The microsensor has the advantages of low cost, low energy consumption, high performance, and compact size. Umesh et al. [58] proposed a micro-electro-mechanical system (MEMS) gas sensor (based on physical changes) to detect the presence of $SF_6$ gas. Using a mixture of carbon nanotubes and graphene-carbon nanotubes as sensing films, the $SF_6$ gas is detected using surface acoustic wave (SAW) structures, which can achieve ppt resolution and operate at room temperature. Gutierrez et al. [59] used non-toxic and sustainable plasma magnesium/magnesium oxide (Mg/MgO) nanoparticles (NPs) to catalyze the degradation of $SF_6$ plasma into harmless magnesium fluoride/magnesium sulfate ($MgF_2$/$MgSO_4$) and produced an effective plasma-enhanced $SF_6$ stoichiometric sensor. Depending on the excitation wavelength, $MgF_2$ is formed when the local surface plasmon resonance (LSPR) is in UV light; $MgSO_4$ is produced in visible light. The platform can also be regenerated by hydrogen plasma in a matter of seconds. The Mg/MgO NPs platform not only absorbs and monitors $SF_6$ in the air but also identifies a new application for plasma catalysis in environmental remediation.

In view of the discussion and analysis in this section above, the performances of these technologies were compared, as shown in Table 2.

**Table 2.** Comparison of detection techniques of sulfur hexafluoride and its decomposition gases.

| Technique | Advantages | Defaults | Gas | Response Time | LOD | Ref. |
|---|---|---|---|---|---|---|
| NDIR | Fast response speed, high sensitivity, high spectral resolution,; remote, and real-time online monitoring | Limited applicability and low sensitivity | $SF_6$ | <25 s | 1 ppmv | [21] |
| QEPAS | Strong anti-interference ability to external noise, small size, and fast response | Signal strength needs to be improved | CO | 3 min | 10 ppbv | [9] |
| | | | $H_2O$ | 1 s | 0.49 ppmv | [22] |
| | | | $SF_6$ | 0.4 s | 4.6 ppbv | [31] |
| CEPAS | High selectivity and sensitivity and wide detection range | High cost and limited scope of application | $SO_2$ | | 35 ppbv | [39] |
| | | | CO | - | 5.1 ppmv | [41] |

**Table 2.** *Cont.*

| Technique | Advantages | Defaults | Gas | Response Time | LOD | Ref. |
|---|---|---|---|---|---|---|
| TIR | High resolution, fast response, large number of detectable elements, high detection accuracy, and strong anti-interference ability | The existence of cross-interference | $SO_2$ $SO_2F_2$ $CS_2$ HF | - - - - | 58.03 55.95 60.72 3 ppmv | [45] [46] |
| TDLAS | Fast response speed, high sensitivity, and high spectral resolution | Severe cross-interference | $SO_2F_2$ $SOF_2$ | | 1.01 ppmv 1.03 ppmv | [47] |
| RS | Enables simultaneous analysis of most gases | The Raman scattering cross-section is relatively small, and the signal is weak | CO $SF_6$ | 20 s | 1.22 ppmv 5.4 ppmv | [10] |
| GC | Can detect most of the gas components | High cost, long response time, large volume, and not suitable for online monitoring or portable testing | $SOF_2$, $SO_2F_2$, $SO_2$, and $SF_4$ | - | µL/L (ppmv) | [53] |
| IMS | Fast response, high sensitivity, low cost, and simple operation | Poor selectivity | $SF_6$ | <1 s | 0.16–0.68 ppmv | [54] |
| Detection tube | Strong pertinence and less gas consumption | Easy to be affected by temperature and humidity, low stability, and high cross-sensitivity | $SO_2$, $H_2S$, CO, $CS_2$, and HF | - | µL/L | [56,57] |
| Nanotechnology and memory sensors | Low cost, low energy consumption, high performance and compact | Temperature and humidity effect is large; Poor selectivity | $SF_6$ | - | ppt level | [58,59] |

## 4. Reliable and Highly Sensitive Gas-Sensitive Materials for Intelligent Monitoring

### 4.1. Carbon Nanotube (CNT) Material

One-dimensional carbon nanotube materials have a large specific surface area and good electrical properties [60]. Carbon nanotubes have good sensing properties for gases [61]. Carbon nanotube-based gas sensors have outstanding characteristics, such as faster response, lower operating temperature, and higher sensitivity [62,63], which have attracted considerable attention. Single-walled carbon nanotubes (SWCNTs) are constructed from graphite sheets, while multi-walled carbon nanotubes (MWCNTs) [64] consist of an array of such nanotubes. The use of CNT-based sensors to detect $SF_6$ decomposition gas can be used as an indicator for monitoring the insulation of power equipment. By doping the sidewalls of carbon nanotubes with metals or nonmetals, their adsorption capacity can be changed. Coupling the doped atoms (s) to a carbon cage can form a mutual region that has a great influence on the adsorption behavior of the resulting gas carbon nanotubes [65]. Yao et al. [66] doped N-3 with carbon nanotubes to study the adsorption of $SOF_2$ and $SO_2$. The results show that the adsorption activity was significantly enhanced after N-3 embedding.

To explore the type and concentration of $SF_6$ decomposition products, based on density functional theory (DFT), Gui et al. [67] modified a nickel-modified carbon nanotube (Ni-CNT) gas sensor to analyze the sensitivity and selectivity of $SF_6$ decomposition components. The results show that the nickel–carbon gas sensor has excellent gas sensing performance. The sensitivity is sorted as follows: $H_2S > SOF_2 > SO_2 > SO_2F_2$. The detection limit of nickel–carbon nanotube gas sensors can reach 1 ppmv.

### 4.2. 2D (Two-Dimensional) Material Sensor Material

As the main component of gas sensors, gas-sensitive materials are currently rich and diverse. After the emergence of graphene, a variety of two-dimensional materials have appeared one after another, such as metal oxides, tellurium, graphene, phosphorene, transition metal disulfides (TMDs), etc., due to the ultra-thin structure, strong surface activity, large specific surface area [68,69], desirable size, power consumption, and optimal operating temperature of the gas sensor being greatly reduced. The researchers also focused on these two-dimensional materials to study their gas sensing properties, as shown in Figure 6.

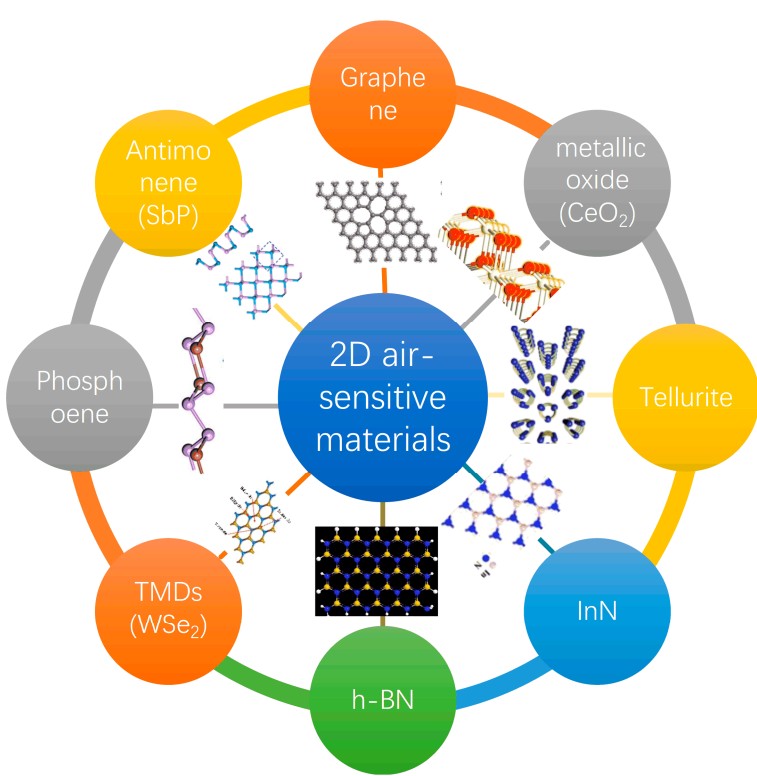

**Figure 6.** Two-dimensional gas-sensitive materials and their geometric structures.

Graphene materials have a unique single atomic layer structure and electronic band structure. Its excellent properties, such as high electron mobility, high thermal conductivity, and large specific surface area, make it an excellent candidate material for gas sensors [70–73]. $Pd_3$ doped with graphene ($Pd_3$-graphene) can chemically adsorb $H_2S$, $SO_2$, $SOF_2$, and $SO_2F_2$ and has the best sensitivity to $SOF_2$ gas. The authors [74] added $N_3$ and Ni to graphene for modification, and they found that the conductivity increased significantly after modifications were made for detecting $H_2S$ and $SO_2$ gases. Conversely, conductivity decreases significantly when $SOF_2$ and $SO_2F_2$ gases are absorbed.

More researchers are setting their sights on graphene-like structures in gas-sensitive materials [75]. h-BN (BN), known as white graphene, is a graphene-like structural material that has better thermal conductivity and thermal stability compared to graphene [76,77]. Moreover, doping metal atoms in BN has higher reaction properties, and the gas greatly enhances sensing capability [78,79]. For example, the Pd metal-doped h-BN monolayer can be used as a nano-sensing material [75]. Another example is the rhodium (Rh) atoms, which are popular doped metal in terms of improving the surface properties of 2D materials, such as graphene, carbon nanotubes, and molybdenum disulfide monolayers due to their high carrier mobility and catalytic interaction with gases [80,81]. Xia et al. [82] studied Rh-doped h-BN (Rh-BN) monolayers to analyze the adsorption of $SF_6$ decomposition gas. In Figure 7a, it can be seen that Rh-BN has a smaller band gap than the original material, electrons are more easily excited, and electrical conductivity is significantly improved. As shown in Figure 7b, under the different temperatures, $SOF_2$, $SO_2F_2$, $SO_2$, and $H_2S$ sensitivity difference is bigger and is advantageous for selective detection. In Figure 7c, the Rh-BN monolayer has good adsorption performance for the four decomposed components, and the negative value of $E_{ad}$ represents the exothermic process. The higher the absolute value, the more easily gas adsorption occurs and the more stable the system. The positive and negative charge transfer corresponds to the characteristics of gas molecules, which will affect the change in electrical conductivity. Compared to Figure 7a, the band gap changes in $H_2S$ and $SO_2$ are greater than those of $SOF_2$ and $SO_2F_2$, indicating that the conductivity changes in $H_2S$ and $SO_2$ after adsorption are greater and the gas sensitivity is higher.

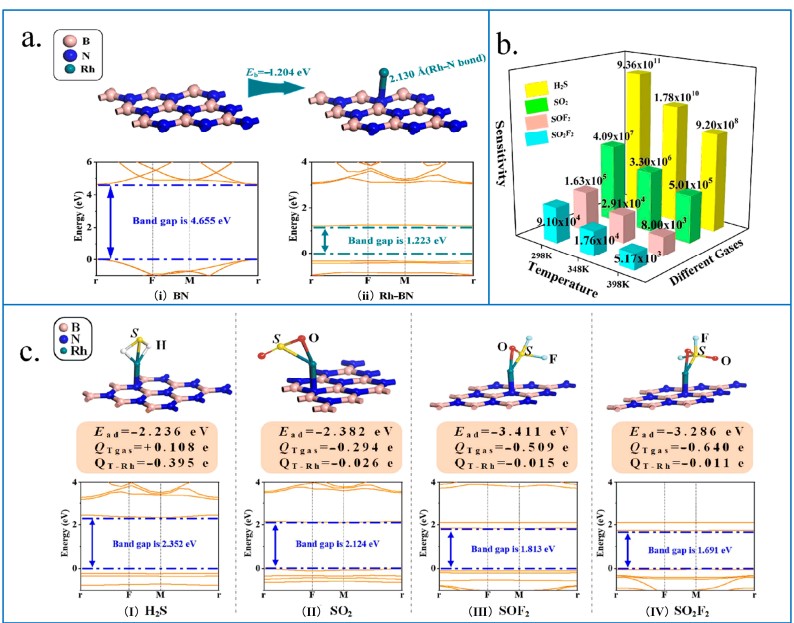

**Figure 7.** (**a**) Band structure of BN (i) and Rh-BN (ii). (**b**) Sensitivity (changes of resistance before and after gas adsorption, S) at different temperatures. (**c**) The adsorption geometry, adsorption energy, charge transfer, and band structure of $H_2S$ (I), $SO_2$ (II), $SOF_2$ (III), and $SO_2F_2$ (IV). Adapted with permission from [81].

Metal oxide (MOX)-based gas sensors are inexpensive and miniaturized and have ease of integration and real-time monitoring. Lu et al. [83] prepared electro-spun ZnO-$SnO_2$ composite nanofiber materials to study their sensing of $H_2S$. The sensor is capable of detecting $H_2S$ at 0.5 ppmv at an optimal temperature of approximately 250 °C at a 50 ppm concentration of $H_2S$ gas. Liu et al. [84] made NiO-modified ZnO nano-flower materials to analyze their gas sensing performance. When the concentrations of $SO_2$, $SOF_2$, and $SO_2F_2$ gases are 100 ppmv, the optimal operating temperature for $SO_2$ detection is 220 °C with a response value of 84.26, and the optimal operating temperature for $SO_2F_2$ and $SOF_2$ detection are both 260 °C. The modified NiO/ZnO composite is about twice as sensitive as pure zinc oxide nano-rods. Yang et al. [85] hydrothermally synthesized cerium oxide nanoparticles ($CeO_2$ NPs) and doped Au, Ag, and Pd as sensing materials. The optimal operating temperature of Au-$CeO_2$ for 50 ppmv $SO_2$ is about 100 °C and Pd-$CeO_2$ has the best response to 50 ppmv $SO_2F_2$ at 250 °C. These studies show that the traditional metal oxide-based gas sensors have problems such as high optimal operating temperature, complex synthesis process, and low sensitivity; so, they are not conducive to online monitoring. The transition metal (oxygen) hydroxides of Co, Ni, and Fe have attracted great interest in gas sensing [86,87] due to their good electrochemical stability. Haq et al. [88] synthesized octahedral $Co_3O_4$-modified $NiSnO_3$ nanofiber materials through a two-step synthesis process. At 50 °C, the optimal sensing response value of the composite is 22.5 when the $SO_2F_2$ concentration is 100 ppmv. This indicates that the material has a high sensing response to $SO_2F_2$ at a relatively low temperature.

Cobalt oxyhydroxide (CoOOH) is a mineral called heterogeneous rock that serves as a precursor for electrode materials such as energy storage, gas sensors, and catalysis [89]. The hydroxyl group on the surface makes the CoOOH more conductive, which is suitable for gas sensing [90]. Opoku et al. [91] performed a functional calculation of dispersion-corrected density and studied the adsorption performance of CoOOH on $SF_6$ decomposition products. The results indicate that the band gap of the gas adsorption systems was all reduced; so, the conductivity and gas sensing performance are improved, among which CoOOH has higher charge transfer and adsorption energy for $SOF_2$. CoOOH flakes can sense 10 ppmv $SF_6$ decomposition gases. Liu [92] incorporated Ni into a ZnO (Ni-ZnO) monolayer to

detect $SO_2$ and $SOF_2$ based on density functional theory. The Ni-ZnO sensing material has a strong chemical adsorption force on the two gases. A significant increase in the band gap of Ni-ZnO monolayer results in a significant decrease in its conductivity, which is sufficient to be detected by electrochemical devices.

In addition to the above, many high-frequency low-dimensional materials have emerged for gas-sensitive materials, such as monolayer tellurium [93], Ru-InN monolayer [94], phosphorene [95], InP3 monolayer [96], monolayer SbP [97], stone-Wales (SW) defective Sb [98], and so on. The band gap, adsorption energy, gas charge transfer, the closest atomic distance to the gas molecules, donor/acceptor characteristics of gas molecules, and band gap after adsorption of gas molecules have been compared in details, as shown in Table 3.

**Table 3.** Band gap, adsorption energy, gas charge transfer, the closest atomic distance to the gas molecules, donor/acceptor characteristics of gas molecules, and band gap after adsorption of gas molecules.

| Materials | $E_g$ (eV) | Gas Molecules | $E_{ad}$ (eV) | $Q_T$ (e) | d (Å) | Donor/Acceptor | $E_g$ (eV) | Refs. |
|---|---|---|---|---|---|---|---|---|
| Rh-BN monolayer | 1.223 | $H_2S$ | −2.236 | 0.108 | - | acceptor | 2.352 | [82] |
| | | $SO_2$ | −2.328 | −0.294 | - | donor | 2.124 | |
| | | $SOF_2$ | −3.411 | −0.509 | - | donor | 1.813 | |
| | | $SO_2F_2$ | −3.286 | −0.640 | - | Donor | 1.691 | |
| Ni-ZnO monolayer | 1.100 | $SO_2$ | −2.38 | −0.183 | 1.89 | donor | 1.515 | [92] |
| | | $SOF_2$ | −2.19 | −0.148 | 1.95 | donor | 1.547 | |
| Monolayer tellurium | 1.233 | HF | −0.314 | −0.016 | 3.919 | donor | 1.214 | [93] |
| | | $SOF_2$ | −0.345 | −0.093 | 3.590 | donor | 1.231 | |
| | | $SO_2$ | −0.382 | −0.158 | 3.278 | donor | 1.071 | |
| | | $SO_2F_2$ | −0.231 | −0.007 | 3.798 | donor | 1.219 | |
| | | $H_2S$ | −0.317 | 0.026 | 3.651 | acceptor | 1.225 | |
| Ru-InN monolayer | 0.359 | $SO_2$ | −2.58 | −0.446 | 2.086 | donor | 0.149 | [94] |
| | | $SOF_2$ | −2.37 | −0.680 | 2.124 | donor | 0.121 | |
| | | $SO_2F_2$ | −2.61 | −0.773 | 2.062 | donor | 0.139 | |
| Phosphorene | - | $SO_2$ | −0.382 | −0.118 | 2.852 | donor | - | [95] |
| | | $H_2S$ | −0.208 | −0.047 | 2.366 | donor | - | |
| InP3 monolayer | 1.018 | $SO_2$ | −1.54 | −0.28 | 2.30 | donor | 1.071 | [96] |
| | | $SOF_2$ | −0.77 | −0.12 | 2.73 | donor | 0.983 | |
| | | $SO_2F_2$ | −0.31 | −0.05 | 3.51 | acceptor | 1.017 | |
| monolayer SbP | 0.28 | $H_2S$ | −0.269 | −0.013 | 3.608 | donor | 1.42 | [97] |
| | | $SO_2$ | −0.541 | −0.232 | 2.974 | donor | 1.76 | |
| | | HF | −0.315 | −0.160 | 2.214 | donor | 1.42 | |
| | | $SOF_2$ | −0.286 | −0.04 | 3.506 | donor | 2.05 | |
| | | $SO_2F_2$ | −0.255 | 0.02 | 3.554 | acceptor | 2.05 | |
| Stone-Wales (SW) defective Sb | - | $H_2S$ | −0.464 | 0.028 | - | acceptor | - | [98] |
| | | $SO_2$ | −0.731 | 0.258 | - | acceptor | - | |
| | | $SOF_2$ | −0.528 | 0.080 | - | acceptor | - | |
| | | $SO_2F_2$ | −0.400 | 0.071 | - | acceptor | - | |

Tellurium-two-dimensional elemental materials, due to their mild band gap, excellent environmental stability, and high room temperature carrier mobility, are widely attractive [99,100]. The excellent properties described above indicate that tellurium can be used in gas sensing directly [101,102]. Cui et al. [93] studied the characteristics of decomposition products on monolayer tellurium based on density functional theory (DFT) calculations. The results are shown in Table 3, and the monolayer telluride has good adsorption and charge transfer capacity for the above components, especially $SO_2$ and $SOF_2$.

Telluride is a 2D nanostructured material that can be synthesized in a variety of ways [103]. Wang et al. [104] designed three types of tellurium-based gas sensors, including ohm-contact monolithic (RCTF), non-ohm-contact monolithic (DCTF), and thermally functioning droplet (HTF) sensors, to study the sensing performance by using the characteristics of tellurium with phosphorus-like fold structure, good air stability, and high gas

sensitivity. RCTF sensors are highly sensitive and selective to hydrogen sulfide, but like most 2D materials, their recovery is slow. Compared to RCTF sensors, DCTF sensors offer significant improvements in selectivity, sensitivity, and recovery speed. The HTF sensor shows that the short thermal pulse is very effective in improving the recovery process of the gas sensor, and the recovery speed is the fastest. Non-ohmic contact between the gold electrode and the tellurium sheet can effectively improve gas sensing performance.

Doping the transition metal (TMs) up to the InN can effectively control the electronic behavior of the InN monolayers, indicating that the doping system can significantly improve chemical activity and electron mobility [105]. Cui et al. [94] investigated the application of doping Ru metal into the InN monolayer (Ru-InN). The results (Table 3) show that Ru-InN monolayer has strong adsorption behavior of $SO_2$, $SOF_2$, and $SO_2F_2$ gases and can identify types of gases, indicating strong chemical interaction.

Transition metal dihalides (TMDs) have received great attention as sensor materials due to their high detection capability, low power consumption, and unique electrical properties [106]. Xu et al. [107] selected the transition metal Pt-doped $WSe_2$ monolayer and analyzed the adsorption of the decomposition gas on the $Pt_n$-$WSe_2$ ($n$ = 1, 2, 3) monolayer based on first principles theory. Pt-$WSe_2$ and $Pt_2$-$WSe_2$ enhanced the adsorption activity of $SO_2$ and $SOF_2$, but both Pt/$Pt_2$-$WSe_2$ reduced the adsorption of $SO_2F_2$. The adsorption performance of $Pt_3$-$WSe_2$ on the above three gas molecules was improved. Wang et al. [108] studied the absorption performance of S and Se in the Janus MoSSe monolayer on four decomposition gases. In Figure 8a, it can be seen that the S layer and Se layer have higher sensitivity and selectivity to $SO_2$ gas, the Se layer is more sensitive than the S layer, and the Se layer has higher adsorption energy and charge transfer for the four decomposition products than the S layer. Figure 8b shows that at high temperatures, the desorption recovery time of $SO_2$ at the S and Se layers is close.

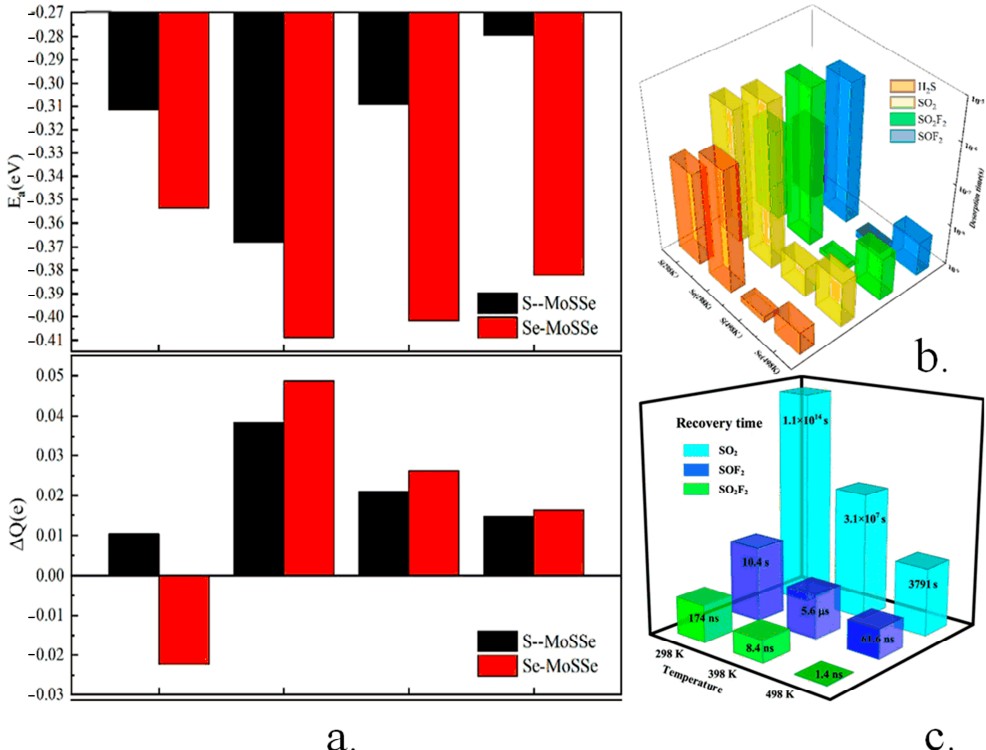

**Figure 8.** (**a**) Absorption energy (eV) and charge transfer (Δq) of $H_2S$, $SO_2$, $SO_2F_2$, and $SOF_2$ in the S and Se layers. (**b**) Comparison of gas desorption times (s) on the S layer and the Se layer at two temperatures (298 and 498 k) [103]. (**c**) Recovery time (s) required for gas desorption on the $InP_3$ monolayer. Adapted with permission from [91].

Phosphate has stronger sensing properties compared to graphene and molybdenum disulfide. Yang et al. [95] studied the possibility of phosphate in $SF_6$ decomposition gas sensors. The results (Table 3) show that the adsorption energy of $SO_2$ by phosphate is significantly higher than that of $H_2S$, and $SO_2$ molecules can be easily adsorbed to the phosphorus surface for desorption. The closest atomic distance between the gas and phosphate is greater than the sum of the covalent radii of the corresponding atoms, which shows that the adsorption of these three gases is physical adsorption.

Recently, a series of 2D metal phosphides, including $InP_3$ [109], $GeP_3$ [110], and $CaP_3$ [111], were found to have high carrier mobility. In particular, it has been reported that the $InP_3$ monolayer has high chemical stability, excellent indirect semiconductor performance, and good optical performance [109], which makes it a novel nanomaterial for applications. Qin et al. [96] investigated the adsorption properties of the $InP_3$ monolayer for $SO_2$, $SOF_2$, and $SO_2F_2$. The result of the adsorption performance (Table 3) for the three gases is $SO_2$> $SOF_2$> $SO_2F_2$. After gas adsorption, the electronic behavior of the $InP_3$ monolayer will be changed, resulting in the redistribution of electrons and a change in band gap and conductivity. In Figure 8c, the desorption time of $SO_2F_2$ is very short at room temperature, indicating the weak interaction between $SO_2F_2$ and $InP_3$. The $InP_3$ monolayer is a good sensor material for $SO_2$ and $SOF_2$ detection. Detectable response and ideal desorption characteristics allow $InP_3$ monolayers to operate at room temperature.

Antimonene is a new two-dimensional (2D) material, with higher air stability, higher carrier mobility, wider adjustable band gap, and a more curved structure than phosphorene [112,113]. It is also considered an ideal material (Table 3) and showed better gas transduction of SbP to $SO_2$ than other gas molecule monolayers. Orbital hybridization is generated by the P orbitals of S and O atoms and the P orbitals of Sb and P atoms, which allows for strong interactions between SbP and $SO_2$. After comparing and analyzing the coefficients of different substrates, SbP is suitable for the sensing of $SO_2$. Antimonene inevitably introduces point defects during the synthesis process. As a typical point defect, the Stone-Wales (SW) defect reduces the structural symmetry and band gap of the antimonene. Antimonene is more likely to form software defects than silicene and graphene. Wang et al. [98] investigated the sensing properties of antimonylene by considering SW defects. Table 3 shows that the sensitivity and selectivity of antimonacene to $SO_2$ are higher than other decomposition products. SW defects make antimonenes more stable and able to withstand interference under harsh conditions.

## 5. Conclusions and Outlook

$SF_6$ leakage monitoring, decomposition product diagnosis, and detection technology are three important components for accurately and timely judging the insulation state of electrical equipment and ensuring the reliability and safety of the power system. Efficient, reliable, and fast monitoring and diagnostic technology can reduce the workload of staff and increase labor efficiency. In this work, the decomposition mechanism of $SF_6$ in the insulation equipment and the reasons affecting the type and concentration of products were summarized, and various detection techniques for $SF_6$ decomposition by-products were introduced to analyze the insulation condition. The trend analysis of sensitivity and emission concentration by carbon nanotube materials and two-dimensional nanomaterials for different decomposition products were summarized, which provided theoretical support for sensing materials related to online monitoring technology.

The content of $SF_6$ and its decomposition products is closely related to the purity of $SF_6$ itself and the surrounding environmental conditions. It is of great significance to understand the reaction and monitor it for the maintenance and safe operation of the equipment. The existing detection methods, such as large-scale equipment inspection, are expensive and complex to operate. Although the cost of miniaturized equipment is low, its detection accuracy is low and can easily be affected by the environment; so, it is difficult to realize real-time monitoring. Gas-sensitive materials are the core components of monitoring equipment, and the integration of the equipment has become its primary

problem such that it needs to be combined with other online monitoring devices, and the monitoring gas range is limited. Gas-sensitive materials can be combined with prior art in a variety of ways. Using gas-sensitive materials as samples or components, the spectral changes under different gases are analyzed. The gas-sensitive materials are integrated into the optical path, and the sensitive response of the gas-sensitive material to a specific gas is used to enhance the signal strength or selectivity of the infrared spectrum. Combined with the change in electrical properties of gas-sensitive materials and the spectral analysis of prior art, a model was built to improve the accuracy of detection.

Due to the existence of various factors affecting the decomposition state of $SF_6$ and the interference between products, there is no established online monitoring method to detect multiple decomposition components simultaneously. It is a promising detection method for the analysis of $SF_6$ and its by-products using a combination of spectroscopy and chemical sensors by machine learning algorithms plus artificial synapses, which can detect multiple components, has strong anti-interference performance with high detection accuracy, and can be commercialized in the future. Artificial intelligence gas monitoring is an important direction of gas monitoring of power equipment for the future. Intelligent inspection can be realized by remote control, with high safety and high efficiency. Real-time data transmission can be carried out during inspection, and the monitoring range is wide. However, the improvement of its environmental resistance and adaptability (high temperature, high pressure, and toxicity), battery life, and data accuracy needs more attention.

**Author Contributions:** Writing—original draft preparation, L.Y.; writing—review, S.W.; supervision, C.C.; validation, Q.Z.; discussion, R.S.; conceptualization, methodology, and funding, Y.H. All authors have read and agreed to the published version of the manuscript.

**Funding:** This research was funded by the National Key R&D Program of China (No. 2023YFC3707201), the National Natural Science Foundation of China (No. 52320105003), and the Informatization Plan of Chinese Academy of Sciences (Grant No. CAS-WX2023PY-0103).

**Institutional Review Board Statement:** Not applicable.

**Informed Consent Statement:** Not applicable.

**Data Availability Statement:** Not applicable.

**Acknowledgments:** The authors thank Hui Huang, Qingqing Shi, Yunhao Cai, Da Zhao, and Yang Wang of the University of Chinese Academy of Sciences for the helpful discussions of this work.

**Conflicts of Interest:** The authors declare no conflicts of interest.

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
