# Peer review of "Monitoring and Leak Diagnostics of Sulfur Hexafluoride and Decomposition Gases from Power Equipment for the Reliability and Safety of Power Grid Operation"

_applsci, doi:10.3390/app14093844_

Round 1

Reviewer 1 Report

Comments and Suggestions for Authors

Dear author(s)!

The manuscript is well organized and well written. However, the following few issues should be taken into consideration.

1) Some abbreviations are not deciphered at the first mention, which makes it difficult to understand the material: QEPAS; CEPAS (Figure 2)

2) From the material presented, it is not entirely clear how the proposed method of technical diagnostics and maintenance can be integrated into the existing equipment monitoring system.

3) There is no quantitative data on the effectiveness in anti-interference performance with high detection (how accurate are they?) and the accuracy of gas leak detection (quantitative data should be clarified). The parameters for assessing reliability and safety, which are presented as advantages of the proposed method, are also unclear.

4) Figures need to be improved in quality

Comments on the Quality of English Language

Minor editing of English language required

Reviewer 2 Report

Comments and Suggestions for Authors

With this article, the authors give a review of different methods and technologies to monitor the leakage of SF6 decomposed gas from electrical equipment. The article is well written and organized and perfectly in the scope of Applied Sciences.

The context is briefly described in the introduction explaining why this monitoring is crucial for electric industries. In the 1st part, they explain the decomposition mechanism of SF6 according ti the type of discharges.

The second part deals with different methods for a direct monitoring of SF6 and decomposition products: IR spectroscopies, Chromatography and Mass spectrometry. I found this section very pedagogical and very exhaustive but to improve the article I suggest to add a Table comparing the advantages and defaults of each techniques  to monitor the leakage of SF6 decomposed gas from electrical equipment. In this table it will be interesting to have a comparison of the limits of detection and the time responses (with the same units) and the ability to detect the different gases (SF6 and main products).

The third part gives an overview of reliable and highly sensitive materials for this gas monitoring. I found that this part is less pedagogical compared to the previous one. I suggest to the authors to give some elements to explain the figures 6 and 7 and also the Table 2. How adsorption energy, gas charge transfer, atomic distances, donor/acceptor characteristics and band gap influence the sensitivity of the material? Moreover the readability of both Fig. 6 & 7 must be improved: the legend of the axes are to small, some units and labelling are missing (e.g: unit of sensitivity in 6b; legend of the different column bars for each molecules in 7a...). 

The conclusion leaves the reader a little disappointed. It could be improved by explaining how to combine the best monitoring technique with the best adsorbent material. Finally, no reference is given to prospects using AI applicable to the problem addressed in the article.

Some minor corrections:

"tunable" instead of "tuneable"

"Beer-Lambert Law" instead of "Bill's-Lambert Law"

"ln" instead "lg" in Eq.1

All the LOD should be given in ppmv/ppbv units (not like as example in l.203)

cm-1 unit should be added for the absorption coefficients

l.374 "sensitive" instead of "ensitive"

l.447 "2" in index

Reviewer 3 Report

Comments and Suggestions for Authors

The sentences in line 20 to 22 are not clear enough. The decomposed gases inside the devices decrease the insulating properties but the leakage is of importance because of the toxicity. The leakage of SF6 is also decreasing the insulating properties. The collapse of the electric system is also not directly bound to the presence of the gases, but the defect of the device.

In line 34: I do not understand how SF6 can be combustible at room temperature when the it is nonflammable.

In line 35: insulation nonconductive should be replaced either for insulating or for non-conductive.

In line 38: insulated should be insulating

At the end of line 105: content in the equipment (the word of is not needed)

In line 144 and 147: the abbreviation TDLAS stands for two different explanation.

In line 156: The referred article [20] does not prove the statement given in that section. The article is about the sensing of gases and not about sensing the leakage. In the article cited the method is with sampling the gas, which means that it is possibly not a measurement during operation and so possibly not appropriate for leakage detection.

In line 202: The sentence is possibly malformed. The detection of laser source is stated which I do not really understand.

In line 217: humidity content of the monitoring equipment... possibly the monitored equipment

In line 239: carbon monoxide should be carbon-monoxide

In line 244, 263, 289: The unit of wavelength is not 1/cm [cm^(-1) as written in the text]! The referred figure contains wavenumber which is related to frequency not wavelength! Either correct the unit (and also the numbers) or change wavelength for wavenumber!

In line 253: instead of "rich detection elements" I would write large number of detectable elements.

In line 279: multiplex laser multiplexing? I suppose laser multiplexing is usually required.

In line 294: "Raman spectroscopy as one of the spectroscopic techniques can be used for gas sensing." should be changed for: Raman spectroscopy is one of the spectroscopic techniques which can be used for gas sensing.

In line 302: As I know light can not be stored... In that case also other electromagnetic waves could be possibly stored (like 50 Hz electricity) which would solve a lot of problems. In laser devices the light is amplified with mirrors not stored.

In line 340: "The first one is to detect changes of chemical."  In my opinion something is missing. Maybe the changes of chemical composition can be detected.

Line 374: in the title gas-ensitive is possibly gas-sensitive

Line 396: "2D material sensor material" To many materials or reformat to be clear what this means.

Line 397: "gas-sensitiv materials currently have a rich variety of gas-sensitive materials" I do not understand what was meant by this sentence.

Line 451: "have attracted greate interest in gas-sensitive" the last word should be possibly gas-sensing.

In Table 2: The word doner occurs multiple times. It should be possibly donor!

Line 523: "which make it a novel as a novel nanomaterial" Should possibly be: which makes it a novel nanomaterial.

space between number and unit in lines 206, 226, 289, 290, 291,

Reviewer 4 Report

Comments and Suggestions for Authors
